# Exploring Chinese and Korean American Teachers’ Perceptions of Their Cultural Identity as Assets and Barriers

**DOI:** 10.3390/bs13120969

**Published:** 2023-11-25

**Authors:** Jiayi Wang, KeAysia Aiyanna Lana Jackson, Eui Kyung Kim, Kevin Han

**Affiliations:** 1School Psychology Program, Education and Social Transformation, Counseling and Educational Psychology Department, College of Health, New Mexico State University, Las Cruces, NM 88003, USA; 2School Psychology Program, Graduate School of Education, University of California, Riverside, Riverside, CA 92521, USA

**Keywords:** Chinese American teachers, Korean American teachers, assets and barriers in teaching, diversity

## Abstract

The representation of Asian American teachers in schools is a unique asset in promoting diversity and advocating for social justice. However, they also face various barriers that negatively affect their work experiences. The diverse nature of the Asian American population is likely to yield different assets and barriers among teachers from different ethnic groups. To gain insight into how Chinese and Korean American teachers perceive their cultural identity in their teaching practices, we utilized a qualitative approach by conducting semi-structured interviews with nine teachers who identified as Chinese and/or Korean American in K-12 public schools in California. Findings based on thematic analyses of the interview data revealed several assets, including empathy and understanding towards minority groups, more exposure to diversity, increased representation, and shared language skills. However, there were also identified barriers, including internalized model minority beliefs, cultural expectations of being passive observers, internalized racism, an overemphasis on education, and a savior complex. Based on our findings, we provide practical suggestions for recruiting and retaining Chinese and Korean American teachers in schools.

## 1. Introduction

The student population enrolled in public schools in the United States (U.S.) is becoming increasingly diverse. Specifically, the percentage of White students decreased from 54% to 46% from 2009 to 2020, and it is projected to drop even further to 43% by 2030 [1]. This means that more than half of the students in public schools are now students of color. On the contrary, the teacher population remains predominantly White, with 79% of public school teachers being White, followed by Hispanic (9%), Black (7%), Asian (2%), and mixed race [2] during the 2017–2018 school year. Although factors such as ethnicity, socioeconomic status, and community membership can complicate the idea of a cultural match between teachers and students [3], having more teachers of color is nonetheless instrumental for students, other teachers, and schools. Research has shown that teachers of color play a vital role in supporting marginalized students (e.g., [4]), promoting culturally responsive teaching among colleagues [5] and advancing social justice in the school system [6].

When discussing diversity in schools, Asian American teachers are often overlooked [7]. Despite Asian students comprising 5% of the student population in 2020, only 2% of public school teachers were Asian American in 2017–2018 [1,2]. The lack of Asian American teachers may perpetuate racial stereotypes and impede the development of empathy towards diversity among students [8]. Kim and Cooc [7,8] conducted two meta-analyses to examine the scholarly attention given to Asian American Pacific Islanders (AAPI) teachers in U.S. K-12 schools. They identified 50 articles that focused on AAPI teachers, with six published between 1990 and 1999, 23 published between 2000 and 2012, and 21 published between 2010 and 2018. Despite an upward trend of studies on AAPI teachers in recent decades, there is still a lack of literature on this specific group of teachers. Moreover, among the 50 articles identified, only a few articles focused on a specific ethnic group under AAPI. Specifically, Sheets and Chew [9] interviewed teachers from diverse backgrounds, with a focus on Chinese American teachers’ perception of the multicultural course during the teacher preparation program. Choi [10] explored the working experiences as the racial minority of two Korean American social studies teachers in high schools. The limited findings of ethnic-specific studies are aligned with a decade review [11] of the Asian American Journal of Psychology (AAJP). Among the 319 articles in AAJP published from 2010 to 2019, the majority of them (*n* = 239) included aggregated or multiple Asian ethnicities. In summary, it is important to have more studies for the Asian American population to increase understanding and representation. Meanwhile, it is meaningful to shed light on the perceptions and experiences of specific subethnic groups among the Asian population considering the differences among ethnic cultures. Thus, this study aims to understand Asian American teachers’ perceptions of the assets and barriers of their cultural identity, with a specific focus on Chinese and/or Korean American teachers.

### 1.1. Assets Brought by Asian American Teachers

Asian American teachers bring valuable strengths to their workplace, serving as cultural role models (e.g., [3,12]), facilitators of culturally responsive teaching (e.g., [13]), and advocates for social justice (e.g., [14]). As cultural role models, they inspire students to strive for academic success and explore diverse career options [3]. Moreover, many Asian American teachers mentor Asian students, helping them navigate negative perceptions about their race and develop a positive racial identity [12]. By educating about Asian culture and identity, their influence extends beyond Asian students, promoting cultural exposure and awareness and reducing racial stereotypes and biases among all students. Through sharing personal backgrounds, Asian American teachers create an inclusive and diverse environment where students are open to sharing and accepting different backgrounds and identities [12].

Research has shown that implementing culturally responsive teaching has a positive impact on student engagement and peer relationships [15,16,17]. Many Asian American teachers play a crucial role in promoting culturally responsive teaching in their classrooms. Specifically, they are conscious of their curricular choices, such as decentering European culture and adding more global perspectives in their lessons. In addition, they create an inclusive classroom environment by having pictures of leaders from different countries and racial/ethnic groups [13]. Some teachers also make an effort to learn more about their students’ backgrounds to facilitate culturally responsive teaching and foster a harmonious classroom [18].

Finally, Asian American teachers can be vital in advocating for social justice practices in schools. Having personally faced racial discrimination during their own schooling, they possess a deep sense of racial and cultural consciousness. Furthermore, their experiences have led them to reject labels such as “model minority” or “White proximity” and embrace their identity as people of color [14]. Consequently, these teachers often encourage their students to engage in critical thinking and discussions on social issues while promoting a safe learning environment to address these sensitive topics [14]. Some Asian American teachers go beyond classroom discussions and motivate their students to take action and make a change. For example, an Asian American teacher in Kim and Hsieh [13]’s study assisted students in drafting a letter of complaint to a testing company about the limited options provided for racial and gender categories on the examinee demographic page.

### 1.2. Barriers Faced by Asian American Teachers

Although they bring unique assets to their classrooms and schools, Asian American teachers also encounter similar barriers to the broader Asian population in the U.S. These challenges include (a) being perceived as foreigners or outsiders, regardless of their duration of residency in the country (e.g., [12]), (b) the “yellow peril” ideology, which views Asian people as a threat to Western civilization and has been aggravated and manifested by discriminatory immigration policies and anti-Asian hate crimes during the pandemic (e.g., [19]), (c) the model minority myth that portrays all Asian people as successful achievers, ignoring the challenges many of them face (e.g., [20]), and (d) “ethnic lumping” which considers all Asian people as a homogenous group, disregarding their diversity in terms of ethnicity, gender, class, and sexual orientation (e.g., [12]).

Studies show that the challenges faced by the Asian community are also prevalent in schools. Kim and Hsieh [13] interviewed 42 Asian K-12 teachers and found that they often encountered Asian stereotypes and the model minority myth from their colleagues and students. They reported often being labeled as “nerdy” and “hard-working” and wanting to be perceived as multidimensional. Unfortunately, these labels led them to question the value of their presence in schools. While some participants believed that hiring Black teachers could challenge negative stereotypes against the Black population, they were concerned that hiring more Asian American teachers would reinforce stereotypes that Asians are only “nerdy” and “smart.” Additionally, Asian American teachers faced oppression in the workplace, such as the perception that they were submissive, which limited their ability to speak up or challenge the status quo. They often felt isolated and invisible in their preservice programs and later career settings due to significant underrepresentation in teaching education and practice. This lack of representation can lead to frustration for Asian American teachers who may feel like they are not being set up for success [13]. This is particularly true when it comes to educational leadership positions, where White men hold the majority of roles.

A study conducted by Endo [21] interviewed 10 Asian American K-12 teachers who reported experiencing various microaggressions in schools. They often felt like they were utilized as a token person of color or cultural expert and not seen as fully “American.” Additionally, they were asked to prepare Asian food or speak on cultural facts regardless of their familiarity with the culture. Other studies [22,23] found that Asian American teachers sometimes faced language barriers, which led to doubts about their teaching abilities and competence due to their accent or proficiency in English. Even when working with students from the same racial background, these teachers were sometimes met with suspicion regarding their cultural backgrounds and even accused of sounding “White” [3].

### 1.3. Theoretical Framework

This study adopted the Asian Critical (AsianCrit) framework [24] to guide the exploration and analysis of the stories shared by participants. Drawing upon the critical race theory, AsianCrit is described as “a perspective that outlines a unique set of tenets that are designed to provide a useful analytic framework for examining and understanding the ways that racism affects Asian Americans in the United States” [24]. There are seven tenets in AsianCrit, including (a) Asianization, (b) Transnational Contexts, (c) (Re)Constructive History, (d) Strategic (Anti)Essentialism, (e) Intersectionality, (f) Story, Theory, and Praxis, and (g) Commitment to Social Justice. This study will focus on the following four specific tenets that are relevant to Asian American teachers’ experiences and perceptions of their cultural identity in schools.

Asianization. This tenet pertains to the way Asian Americans are racialized by society, including being viewed as model minorities, honorary Whites, and yellow perils, and being subject to ethic lumping. Such practices have a prevailing impact on the experiences and identities of Asian Americans in the U.S. In line with this tenet, we aim to examine how Asianization affects Asian American teachers, how these racialized ideologies are internalized and influence their self-perception, and how these perceptions can hinder their teaching practice.

Intersectionality. This tenet considers how race and ethnicity intersect with other social identities, such as gender, age, class, sexual orientation, ability, and immigration status. Asian American’s experiences may not only be influenced by their racial identity, but also by the interplay of other identities. For example, someone who identifies as both Asian and a gender minority may face greater challenges in society. Thus, we plan to examine how the diverse social identities of Asian American teachers influence their teaching practice.

Story, Theory, and Praxis. This tenet emphasizes the connection between individual stories, theoretical work, and practice in comprehending the experiences of Asian Americans. For example, stories can inform and enrich theoretical frameworks, while theories can provide guidance for practical frameworks, and practices can update and refine theories. It is important to provide a platform for Asian Americans to express themselves, and it is equally crucial to value their views and knowledge to create a better understanding of the Asian American community and their culture. We aim to closely listen to and analyze every word spoken by our interview participants. Furthermore, we will use the data gathered in this study to establish concrete recommendations for real-world practices.

Commitment to Social Justice. This tenet highlights the ultimate goal of this framework, which is to advocate for social justice and eliminate different forms of oppression. Our study aims to shed more light on Asian American teachers’ experiences in social justice practice and how their identity can aid their advocacy work. Additionally, we aim to discover practical ways to support Asian American teachers in preventing workplace oppression.

### 1.4. Statement of Problem

Within the limited literature on Asian American teachers, much of the focus was on the barriers and challenges they face (e.g., [21]), with little attention paid to the unique assets they bring to the table. Additionally, there is a lack of literature to understand the assets and barriers for teachers who identify as Chinese or Korean American specifically. Our first research goal is to delve deeper into Chinese and Korean American teachers’ perceptions of their cultural identity as assets to the school and in their teaching practice. We also recognize the external factors that may contribute to challenges, such as racialized assumptions and treatments from colleagues and administrators (e.g., [19]). Examining these contextual factors aligns with AsianCrit since it is crucial in understanding the root causes of racism that shape Asian Americans’ experience in society. We also aim to understand how Chinese and Korean American teachers internally view their cultural identity as either an asset or a barrier to their teaching. Internationalized racism, where individuals from marginalized backgrounds blindly follow the dominant culture while devaluing their own culture, can be detrimental to individual identity, self-esteem, and daily work functioning [25,26]. Thus, this study aims to answer the following research questions:(1)How do Chinese and Korean American teachers perceive their cultural identity as an asset in their teaching practice?(2)How do Chinese and Korean American teachers perceive their cultural identity as a barrier in their teaching practice?

## 2. Method

To answer the research questions and gain insights into Chinese and Korean American teachers’ perceptions, we used a qualitative research methodology comprising semi-structured interviews with each teacher participant. Individual interviews were selected to create an atmosphere of trust and openness that would encourage participants to share their personal narratives and sensitive information [27].

### 2.1. Participants

Participants in this study were recruited from another mixed-method study of Asian American teachers in California. To be eligible for the larger study, all participants had to identify as Asian or Asian American and be a K-12 teacher in California during the pandemic. Among the 12 participants recruited for the study were four Chinese, four Korean, one mixed Chinese/Korean, one mixed Filipina/Latina, one Vietnamese, and one Cambodian. From this group, we chose nine individuals who specifically identified as Chinese American and/or Korean American for this study, excluding individuals whose ethnic identity was mentioned only once. Considering the ethnic diversity of the Asian population, it is important to narrow down ethnic groups to understand their unique characteristics and avoid a pan-Asian approach. Participants chose their own pseudonyms to maintain anonymity. Their demographic characteristics can be found in Table 1.

### 2.2. Procedures

Participant recruitment for the larger mixed-method study included both convenience and snowball sampling. The research team first distributed a recruitment flyer to individual teachers and teaching organizations in California and posted it on social media platforms (i.e., Facebook and Twitter) and in specific Asian American teacher groups (e.g., California Association for Asian and Pacific Education). The flyer provided access to a Qualtrics survey with prescreening questions to determine eligibility for the study. Interview participants were encouraged to forward the study information to anyone they knew who might be interested in the study.

The interviews were conducted over Zoom, with a participant and a researcher present. Zoom interview was selected over the conventional in-person interview because it allows this study to reach more potential participants without regional limitations. They lasted approximately 45 to 60 min in length. A semi-structured interview method was utilized as it allowed for more flexibility and in-depth responses [28]. All the interviewers followed the same set of questions, but had the flexibility to ask more questions based on the participants’ responses. The questions asked ranged from demographic questions (e.g., What is your ethnicity or nationality?), personal background and history in education (e.g., How did you decide to become a teacher?), school characteristics (e.g., What current school population do you work for?), and assets/barriers that Chinese or Korean American teachers perceived of their cultural identity (e.g., Are there things that Chinese or Korean American teachers offer/provide that are different from what White teachers have to offer? How so?).

We recorded all interviews and had them transcribed into written text by a transcription company called REV.com. We then used Dedoose 9.0.107, a qualitative software, to analyze and code the transcripts. After analyzing and interpreting the data, we conducted a member check where one participant reviewed the manuscript and confirmed the interpretation of the quotes.

### 2.3. Data Analysis

Three researchers in this study conducted the interviews and analyzed the data. One of them is a first-generation immigrant from South Korea and a faculty member at an American Psychologists Association (APA)-accredited and National Association of School Psychologists (NASP)-approved school psychology program. The other two researchers are both second-year doctoral students in the same program. One of the student researchers is Black, and the other is a first-generation immigrant from South Korea. The research team utilized the practice of reflexivity prior to the coding process. Reflexivity is an important practice that allows researchers to critically analyze and evaluate their own identities that may influence the research process [29,30] and helps to reduce potential subjectivity and bias, which is common in qualitative research [31]. The team wrote reflexivity statements that highlighted their own identities and potential areas of bias they could have in the subject of Asian and Asian American teachers and education.

This study used reflexive thematic analysis to analyze and code the data collected [32,33]. Throughout the process, the team held multiple meetings to create a codebook consisting of themes and codes. The initial step involved each researcher reading through all the transcripts to familiarize themselves with the interviews. Our coding process included both deductive and inductive analysis. Initially, we grouped responses to answer the two research questions regarding teachers’ perceptions of assets and barriers using the deductive approach. Then, we used the inductive approach to create a set of codes based on participants’ responses. Those codes were then grouped into common themes based on each research question.

To ensure intercoder reliability (ICR), we followed the recommendation of previous studies [34,35] and selected three transcripts, which accounted for 33% of the dataset. The team used the unitization strategy proposed by Campbell et al. [34] to increase ICR across multiple researchers. One coder went through the three transcripts and bracketed portions of the text that needed to be coded. The remaining coders independently worked on coding the selected text, and Cohen’s kappa was calculated to determine ICR. ICR across all coders ranged from 0.93 to 1.00. After achieving adequate ICR, each coder was able to code the rest of the transcripts independently [35].

## 3. Results

The purpose of this study was to explore the viewpoints of Chinese and Korean American teachers about the cultural assets and barriers they face while navigating the school environment. The key themes related to cultural assets and barriers are summarized below and listed in Table 2.

### 3.1. Cultural Assets

#### 3.1.1. Understanding and Empathy toward Minority Groups

Participants revealed that their identity as a marginalized individual helped them better understand and empathize with the challenges faced by students from minority groups. As per some participants, Asian American teachers stood out from White teachers because they had first-hand experience of being part of a racially minoritized group in society. For example, Jasmine, a second-generation Chinese American female teacher, has been teaching for three years at an elementary school. Having personally faced racial discrimination, she feels that she is more aware of what it is like to be a minority in the classroom and the school environment:


*Being a person of color and experiencing just any form of discrimination or prejudice based on how people see me at first glance, I think that is something that white educators cannot fully grasp and cannot understand. It’s kind of what I said before of just you can learn a lot by listening, but there are also only so many things that you can understand by listening when you have not experienced them yourself.*


Olive, who is a 2.5-generation Korean/Chinese American female teacher, teaches students in the third and fourth grades. She has had a diverse education, having studied in both the U.S. and Korea. She further described how her racial background made her more aware of her students’ diverse cultural values.


*I think because Asians are a minority and we had to live a life of being a minority, we notice and understand culture and are more cautious of it because we are the different ones in the culture. And so I’m super hyper aware of different cultures and trying to be more sensitive… And I think Caucasians definitely don’t have to do that because they just never were in a position to do that.*


During interviews, some participants also shared their understanding of family dynamics in many Asian cultures, particularly the emphasis on academic excellence. Olive noted that while high academic expectations may be widespread in Asian cultures, there is also *“a lot of grace and space to grow.”* She understands the pressure some of her Asian students face from their parents and tries to create a balance between academic expectations and their overall well-being.

YL, a Chinese American teacher who identified as 1.5 generation and gender non-binary, shared their school experience with teachers who did not understand their family values. When YL expressed the academic pressure from parents to their White teachers, YL was given an unhelpful solution of ignoring and not listening to their parents. YL is currently teaching in a Cantonese bilingual program that mostly comprises Chinese American students. They believed that being a Chinese American teacher could give students a better perspective and *“a better bond just because they [students] are aware that I understand their background and I also understand where they’re coming from or some hardship that might come with the culture that other people and other teachers might not understand.”*

Another 1.5-generation Korean American female teacher named Cynthia who has five years of teaching experience touched on the intersectionality of race/ethnicity and immigration status. Cynthia was born in Korea and moved to the U.S. with her parents shortly after she was born. She pointed out that she could *“relate to the immigrant struggles like growing up in an immigrant family”* with many students in her class. Cynthia also exhibited a mixed view of her racial/ethnical identity when it came to working with students from the same background:


*If they’re the same racial background, they feel more comfortable with you, so they respect you more. Or because they’re comfortable with you, they don’t respect you that much. They expect more out of you, because they’re like, ‘Oh. You have the same background as me. You should do more for me.’*


#### 3.1.2. Exposure to Diversity

Participants reported that their racial and ethnic identity can help promote diversity among their students. By being present in schools, they can bring attention to ethnic diversity for Asian American students. Their presence also provides an opportunity for non-Asian students to gain a deeper understanding of the complex experiences of Asian Americans, beyond stereotypes and misconceptions. The teachers believed that exposing students to diversity is crucial because it helps them understand that teachers can come from different backgrounds and that not all Asians are the same. To expose them to diversity, these teachers teach an inclusive curriculum and introduce diverse cultural elements to both Asian and non-Asian students. At a surface level, this involves showcasing various aspects of the Asian diaspora, such as customs, holidays, and cuisine. Additionally, exploring cultural identities, understanding cultural values, and addressing racial trauma is necessary for a deeper level of understanding. By doing so, teachers not only disseminate cultural knowledge but also foster intercultural exchanges among students.

For example, Jasmine, a second-generation Chinese American teacher who is one of the few Asian American staff members at her school, noticed that many students had little exposure or understanding of Asian cultures. Thus, Jasmine would talk about her ethnic identity and introduce cultural elements to her students:


*They live in a community where there are not a lot of Asian Americans. Even if there are, there’s not a lot of distinction between different Asian identities, of Chinese, Korean, Hmong, Vietnamese. A lot of our students who are Asian at our school are Hmong or Vietnamese. It becomes important when…I’ll teach my kids about Lunar New Year or I’ll teach them about searching for types of foods that my family likes to eat.*


Jasmine also used her cultural identity as a teaching approach to encourage her students to understand their own identities and learn to talk about those of others. If her students confused her with other Asian staff members or commented on how they looked alike, Jasmine took the opportunity to discuss her Chinese American identity: *“In those cases, my identity as a Chinese American will come up and then I’ll talk about it.”* Furthermore, she said:


*I usually lean on it to help my students understand their own identity. But it’s something that I use more as a teaching tool to respond to a moment, my ethnic identity is not really something that I will teach or talk about unprompted. I kind of spend more on helping my students to understand how to talk about their own and each other’s identities first.*


Phoebe, a new middle school teacher who is a 1.5-generation Korean American female, shared her strategy of using food to introduce her students to various cultural elements. She created opportunities for her students to share their ethnic food with one another, fostering a positive approach towards understanding different cultures. Phoebe also emphasized the importance of teachers understanding their own ethnic identities in encouraging their students to respect and appreciate cultures that are different from theirs.


*Being Korean is important so that they understand to respect not just me, as Korean, but just every other ethnicity out there. For them to be exposed to different skin of color, I think that part is important. I hope they learn that now, since they’re kids.*


Additionally, some teachers found it important to incorporate their own cultural heritage into their classroom curriculum. Cris is a second-generation Korean American female teacher with 20 years of teaching experience. She disclosed that she was very “white-centered,” growing up in a white community, and thus, she initially *“connected more with the whiteness of her schools”* in her early years of teaching. However, she gradually realized that she was *“trying to live up to the reputation that Asian People have in the U.S.”* and started to integrate her own cultural wealth in teaching.


*I do think that there is something very powerful about seeing somebody in a leadership position that looks like you. And it also allowed me to tailor the curriculum a little bit so I could pick out the more Asian-centered literature, especially. That was really hard when I started trying to collect things from my library, there was none. Now there’s a lot, but before there was none and to choose more topics that were inclusive and diverse, that included Asian population specifically Korean was very hard.*


She also confirmed the unique asset of Asian American teachers in teaching about the history of Asian populations in the U.S.


*I think there are a lot of things that Asian teachers teach that would be inappropriate for white people or white teachers to breach… So I do feel like we have the corner of that. We can actually go into those deeper discussions and even in some cases, some of those more difficult topics about the Asian history than white teachers.*


#### 3.1.3. Representation

During the interviews, representation emerged as a prominent theme that was divided into two subthemes: Asian representation (specific nuances for serving as representation for Asian students) and Asian x Female (specific nuances for serving as representation for Asian girls).

Some teachers discussed the importance of being representation for Asian American students specifically. Jasmine, a second-generation Chinese American, aimed to inspire her students to pursue any career they desired, even if it was not culturally expected. She believed that by doing so, her students would gain the competence to pursue their own goals:


*I think it’s important to have representation, right? If we’re talking about Asian American students having Asian American educators, one of the reasons being just kids seeing that they can do any job. There’s a really big stereotype about Asian Americans or Asians going into certain professions. I know my parents didn’t want me to go into the profession that I’m in, because they felt it was unstable or was unsafe, or I wouldn’t make enough money as much money as they want me to. I think it’s important to have representation, to just show Asian American kids that they can just do what they want to do.*


Cris, another teacher who is a second-generation Korean American, highlighted the intersectionality of representation. Her ethnicity and gender combined made Asian female students feel at ease in her presence, to the point where they sometimes referred to her as “mom” by mistake.


*For some reason, I always think for a woman because I identify as a woman, teaching Asian girls has been a very interesting transformation. I never really realized how impactful it is until they come into your classroom, and they are just like, like they call me mom. I mean, they mistakenly call me mother or mom. It’s really interesting, and it’s actually something that I’ve really come to love, just representing and making that a powerful statement for them too. And just having them realize they can do big things and they are not repressed from it. They feel far more comfortable talking to me, I think.*


Phoebe also acknowledged the intersectionality of her race/ethnicity and English language proficiency. Having moved to the U.S. at the age of eight, she was an English language learner and still has an accent, as reported by herself. She felt a deeper connection with students who are also English language learners because they went through similar struggles learning a different language. Phoebe believed that, *“for students, because they hear my accent and because they know that I was also an English learner, I think that builds the connection between (me and) them.”*

#### 3.1.4. Ability to Speak Their Language

Some teachers highlighted the value of having Asian American teachers who can speak the same language as their students. Veon, a second-generation Chinese American female teacher, teaches in a Chinese bilingual program and believes that her ability to speak Cantonese allows her to exchange intracultural experiences and connect with her Chinese students in a unique and meaningful way.

In addition to improving communication with students, having a shared language also makes it easier for teachers to communicate with parents. As a fifth-grade teacher in a Cantonese bilingual program, YL (1.5-generation Chinese American) expressed that they enjoyed being able to communicate with parents in Cantonese and felt that it brought them closer to the family.


*I got into this profession, in this field, this very specific Cantonese bilingual teaching role because I want to not just help the students, but also be a help of the families. Just as simple as a parent-teacher conference, letting them know how their child’s doing. A lot of them work all day, right, and they don’t speak the language, so I think just because we can communicate and understand each other’s culture without a translator in between makes teaching and learning a lot easier.*


Overall, the Chinese and Korean American teachers believed that they could understand and empathize with minority groups, expose students to diverse cultural elements and curriculum, enhance representation for Asian students, and communicate more effectively with their students through a shared language. Through these assets, Chinese and Korean American teachers aimed to promote student success, dispel stereotypes, and foster intercultural exchanges in the school environment.

### 3.2. Cultural Barriers

The Chinese and Korean American teachers emphasized the valuable cultural wealth and knowledge they brought to the school environment. However, they also reflected on certain aspects of their upbringing, and cultural expectations posed challenges for them in navigating the school setting effectively.

#### 3.2.1. Internalized Model Minority

The most discussed cultural barrier among the teachers was the internalized model minority. They often faced the challenge of living up to the model minority myth, which perpetuates the idea in U.S. society that they should be submissive and uncomplaining. This internal pressure can cause stress and difficulty in navigating relationships and inciting changes in the school setting. For example, Jasmine (2-generation Chinese American) described how the internalized model minority myth affected her interactions with White school staff. She described an incident with a senior white teacher who had two more years of teaching than Jasmine. The White teacher was shocked that Jasmine received a contract while she did not and implied that Jasmine was awarded the contract because of her Asian American identity, implying that the district prioritized minority status for employment. Other White coworkers also commented on the unfairness of the district awarding contracts based on ethnicity while sitting with her, leaving her feeling invisible and hurt. According to Jasmine, *“it felt it was kind of about me or it had to do with me, but there wasn’t really any acknowledgment that as they were talking about people of color, that I was a person of color. I very much felt kind of not in that room, or just not noticed that, or I don’t know.”* Under that circumstance, she felt compelled to be *“part of the model minority myth being a good minority”* who was “*not going to stir up trouble or speak up.*” She remained silent in the face of microaggression, which made her feel like the school was a challenging environment.

On a systemic scale, Cris (second-generation Korean American) described the permanency of the internalized model minority myth in Korean culture and how it acted as an agent of white supremacy to maintain the status quo. She expressed her frustration with the pervasive model minority myth in her culture and her feeling of helplessness for students coming from a family strictly expecting their children to behave well and study hard, which fits in with white supremacy.


*It’s really difficult for me to look at my own culture and see that it’s still very pervasive and a very powerful force and it’s not going to go away anytime soon or ever. Because a part of that is interwoven into the Asian culture, at least the Korean culture. Good student, well behaved, good listener. There’s definitely a generational hierarchy. A lot of that fits in perfectly with white supremacy and the white supremacist culture and especially in schools because you’re shaping young people and it’s not just you, it’s an entire family that follows you around and watches you every move. So it’s hard for me to go in and tell people this is actually really bad for you and you should stop doing it out of experience.*


Influenced by the model minority myth herself, Cris described how this internalized model minority myth that Asians respect and the notion to not challenge hierarchy impacted her ability to have effective conversations with administrators about change:


*I think the relationship (with administrator) is complicated because there’s a hierarchy…I struggle with respecting people who are in leadership positions. I have a lot of trouble breaking through that barrier just because I’ve always been told my entire life not to. So when I do eventually say something that is untoward or she doesn’t appreciate, it’s really hard for me to accept it and say, no, I did the right thing… I question everything I say to her and it’s not just because she’s White, it’s also because she’s the person in power.*


#### 3.2.2. Cultural Expectations of Being an Observer

A few teachers reflected on the family and cultural expectations of Asians to be quiet observers, which had an adverse effect on their participation in school as a member of the community. During interviews, multiple teachers admitted that they were reluctant to speak out, fearing that they would go against the cultural expectations of Asians to be quiet, passive, humble, and shy. For example, YL, a 2.5-generation Chinese American, noted that their upbringing taught them to not speak up and led them to *“just sit in the corner and be a fly on the wall and not say anything.”* Similarly, Cris, a second-generation Korean American, further described that these expectations made her take on a quiet “listener” role as a teacher early in her career. However, as she aged, she became increasingly upset with negative practices deteriorating her school and realized that initiating and being an active participant in challenging conversations were necessary for change. This realization came about during equity training when she was 52 years old, and it entirely changed her perspective on the matter:


*Because it’s so ingrained in our history and our culture… to be quiet and actually be more of an observer and a listener than a participant, the quiet participants what I think I would describe it as… I really didn’t think about (her cultural identity affecting her as a teacher) until much, much later in life when I realized it’s easier to see the patterns that are deteriorating your school or your district or your city… I started noticing it and getting more upset by it… Once we started equity training, I started realizing the conversations that were challenging and difficult, it was actually what opened up and entirely new perspective for me. It really did transform my entire teaching. That was just recently.*


Cynthia (1.5-generation Korean American) extended this barrier to job interviews. She expressed admiration for White candidates who were more expressive and articulate.


*I think because Asians, as a culture, you’re told to never speak up and not speak up too much. We’re not used to speaking well, especially in public. We have a hard time. Whereas white folks, they’re very good at speaking, especially in the public setting. For example, an interview, they just know how to talk, really know how to talk.*


#### 3.2.3. Emphasis on Education

One Korean American teacher, Phoebe, found that her upbringing of prioritizing education could be a barrier for her to connect with students. Growing up in a family that placed great emphasis on receiving quality education and excelling academically, Phoebe developed a mindset that emphasized education above all else. Consequently, she tended to assume that students’ families would share similar values and expected students to actively participate in their learning. However, due to the pandemic, many families faced significant challenges, making it increasingly difficult for students to engage in their education while managing other responsibilities. Thus, Phoebe acknowledged that she found it *“a little difficult to relate to the students, with my experience of always putting my education my priority.”*

#### 3.2.4. Savior Complex

Finally, Jasmine (second-generation Chinese American) raised awareness about the “savior complex” that some Asian American teachers, especially in Title I schools, may possess. She explained that some Asian American teachers from higher socioeconomic backgrounds might unknowingly adopt a negative mindset toward their students:


*Asian Americans don’t always realize that we are also very prone to falling into a white savior complex, a savior complex, because I think a lot of, at least East Asian Americans, come from wealthier situations or more middle class or upper middle class. I feel when we teach in Title I schools, I saw this in my teaching program, we just have to have an awareness of having a savior complex. I think sometimes it’s dangerous, because we think that we won’t because we’re not white and it’s called a white savior complex.*


## 4. Discussion

This study confirms previous research on the strengths and challenges of Asian American teachers. In addition, it sheds light on the perspectives of Chinese and Korean American teachers regarding how their cultural identity impacts their teaching, whether it serves as an asset or a barrier.

### 4.1. Assets of Chinese and Korean American Teachers’ Cultural Identity in Teaching

In our study, the majority of participants indicated that having shared experiences, such as experiencing racialized treatment as a minority or coming from an immigrant family with high academic expectations, helped them understand and empathize more with Asian and other minority students. Some teachers also mentioned how race/ethnicity intersects with gender, immigration status, and English proficiency, which helped them connect better with their students. This aligns with previous studies (e.g., [12,36]) that showed how Asian American teachers have faced marginalization themselves could establish trust and rapport with Asian students and their families. Our findings suggest that having more underrepresented teachers who can relate to their students’ backgrounds and experiences can help build positive relationships. Additionally, teachers with cultural knowledge and inclusive teaching skills were seen as valuable assets, as they were able to create learning opportunities that expose students to diverse cultural elements and curricula (e.g., [18]).

The findings of this study also uncovered some unique cultural assets of Chinese and Korean American teachers, which were not mentioned in the previous literature. For example, these teachers not only introduce students to Asian culture, but also highlight the ethnic diversity within Asian culture. This is important as people, including students and school staff, often tend to group all ethnicities together under the umbrella term ‘Asian,’ which can lead to neglect of unique experiences and perspectives of specific groups. Thus, Chinese and Korean American teachers, as well as teachers from other ethnic groups, can incorporate their own values and history into their teaching to enhance a deeper understanding of Asian and ethnic culture among all members of the school. Moreover, their ability to speak languages other than English can further facilitate communication with students and parents, making them valuable assets to the teaching profession.

### 4.2. Barriers of Chinese and Korean American Teachers’ Cultural Identity in Teaching

This research shed light on how cultural identity can be seen as a barrier by teachers. As per the concept of “Asianization” and other studies on Asian American teachers (e.g., [13,20]), many teachers felt pressured to conform to the model minority stereotype. Chinese and Korean American teachers found it difficult to challenge the status quo and push for changes because of cultural pressure to conform to White culture as a model minority, which negatively impacted their experience as teachers. Many participants also reported experiencing internalized racism [25]. For example, several teachers felt that their cultural expectations to be quiet listeners and observers instead of active participants were critical of their own upbringing and were a barrier to their teaching career. These internalized model minority myth and racism put them at a disadvantage in a White-dominant society where assertiveness and advocacy are highly valued. Following these cultural expectations could lead to them being perceived as lacking leadership skills or being less valued as a group member.

Additionally, a participant noted that some Korean American teachers who come from an upper-middle-class background might unconsciously adopt a “white savior complex.” This occurs when they assume they know what is best for their marginalized students and try to rescue them. This behavior may stem from their internalized “model minority” or “white proximity” and can prevent them from building genuine relationships with students. This finding suggests that the internalized model minority myth, combined with privileges in socioeconomic status, can negatively affect how Korean and Chinese American teachers perceive and interact with minority students. Furthermore, the study revealed that some Korean American teachers found it challenging to connect with students and families who did not prioritize education, especially during the pandemic when students were less engaged in learning. This finding highlights the need for Korean American teachers to be mindful of how their values influence their understanding and interaction with students.

## 5. Limitations and Future Research

This study delved into the perspectives of Chinese and Korean American teachers on how their cultural identity could either enhance or hinder their teaching. It is important to acknowledge that this study has some limitations. This study focused on teachers from Chinese and Korean ethnic groups, but it should be noted that there are still variations within and between their cultures. Due to the relatively small sample size in this study, the results reflect in-depth individual experiences and perceptions instead of presenting generalizable characteristics. Future studies are encouraged to increase the sample size to better understand Chinese and Korean American teachers’ working experiences. It would also be informative to examine Asian school staffs other than teachers (e.g., administrators, school psychologists) to understand if they have similar or different experiences from teachers. While it can be challenging to recruit participants with similar backgrounds across various aspects of identity due to the small population within the profession, future studies should strive for improved sample homogeneity to gain a more nuanced understanding. It would be beneficial to further investigate the unique cultural identity of other ethnic Asian American teachers and how it affects their ability and effectiveness as educators. For example, Southeast Asian American teachers, such as Vietnamese and Filipino, may face distinct advantages and challenges that may be similar or different from those experienced by Chinese and Korean Asian American educators. Our study also found that Asian American identity can intersect with other salient identities, such as gender, English proficiency, or immigration status, which warrants further exploration to develop tailored support plans. Furthermore, it is worth noting that this study’s participants were based in California, which has the second-largest Asian American population in the U.S. [37]. Asian American teachers in areas with smaller Asian American populations may experience their cultural identity differently in school settings. Conducting additional research on diverse groups of Asian American teachers would deepen our understanding of their perceptions of their own cultural identity.

This study takes a comprehensive approach to identifying both assets and barriers that Chinese and Korean American teachers face as a minority group in schools. Instead of exclusively focusing on their shortcomings, future research on Asian American teachers should shift its attention to their strengths. While it is critical to explore strategies to diminish barriers, such as providing aide to Asian American teachers who struggle with self-advocacy due to the internalized model minority myth, scholars should also explore the strengths and assets these teachers bring to the classroom. For example, there is a need for more research on how Asian American teachers employ shared experience, which was the most frequently mentioned strength among our participants, to enhance their students’ learning [7] and empower minority students in schools. This asset-based approach can provide a better understanding of the experiences of Asian American teachers within the teaching profession.

## 6. Practical Implications

The aim of this research was to investigate the viewpoints of Chinese and Korean American teachers on how their cultural identity impacts their teaching experiences, whether positively or negatively. The valuable insights gained from their perspectives can offer practical implications to enhance teacher recruitment and retention, leading to the representation of more Asian American teachers and their strengths in schools. To recruit more Asian American teachers, school administrators must assess their behaviors in the context of their unique cultural background and avoid comparing them with the expectations of White culture. For example, in Chinese and Korean cultures, it is customary to show respect by listening more and talking less, which may be misinterpreted as a lack of competence as educators. If administrators are unaware of this cultural difference, they may develop an unfavorable impression of Chinese or Korean American applicants, which could disadvantage them in job applications or field placements. Chinese and Korean American teachers can promote ethnic diversity awareness and create more inclusive content in teaching. Schools are advised to listen to the inputs from their teachers and offer them resources to enrich the curriculum available at school.

To promote inclusivity in the workplace for Chinese and Korean American teachers, school staff should take the following steps: (a) educate themselves on the experiences of Chinese and Korean American teachers related to their racial and ethnic identity, (b) reflect on their own biases and stereotypes towards Chinese and Korean American teachers, and (c) make collaborative efforts to ensure that all teachers feel valued and heard in the workplace. Ongoing training, reminders, and practice are necessary to address ingrained racial biases and promote cultural humility, sensitivity, and awareness and should target all school staff. While many Chinese and Korean American teachers considered their shared marginalized experience as an asset for building relationships with students, some expressed uncertainty in supporting students from different backgrounds, such as those from varying socioeconomic backgrounds. Continued training is crucial for both Chinese and Korean American teachers, as well as those from other backgrounds, to improve their interactions with each other and foster positive relationships with their students. In a society where racial biases are prevalent, some Chinese and Korean American teachers may internalize stereotypical perceptions and become critical of their own cultural values. Peer discussions can help them realize that internalized racism is a common experience and provide a platform to process complex feelings together [38].

## Figures and Tables

**Table 1 behavsci-13-00969-t001:** Demographic characteristics of participants.

Participant Pseudonym	School Type	Gender	Ethnic Group	Generation in the U.S.	Years of Teaching
Veon	Elementary	Female	Chinese	2	3
Charlie	Middle	Male	Chinese	2.5	6
Jasmine	Elementary	Female	Chinese	2	3
YL	Elementary	Non-binary	Chinese	1.5	3
Olive	Elementary	Female	Chinese/Korean	2.5	1
Hailey	Middle	Female	Korean	2	3
Cris	Elementary	Female	Korean	2	20
Phoebe	Middle	Female	Korean	1.5	1
Cynthia	Elementary	Female	Korean	1.5	5

**Table 2 behavsci-13-00969-t002:** Themes summary.

Research Question	Theme	Description
Cultural Assets	Understanding and Empathy toward Minority Groups	The shared experience allows Asian teachers to understand and empathize with the challenges faced by students from minority groups.
	Exposure to Diversity	Asian teachers’ racial and ethnic identity can help promote diversity among their students.
	Representation	Asian teachers can serve as representation for Asian students and Asian girls.
	Ability to Speak Their Language	The ability to speak the shared language with students and parents enables better communication and relationships.
Cultural Barriers	Internalized Model Minority	Asian teachers feel the pressure to live up to the model minority myth in school settings.
	Cultural Expectations of Being an Observer	Expectation from others to be quiet makes Asian teachers reluctant to speak up.
	Emphasis on Education	Family upbringing of prioritizing education makes it difficult to relate to students from a family not emphasizing education.
	Savior Complex	Asian teachers who want to help students unknowingly adopt a negative mindset toward their students.

## Data Availability

The data presented in this study are available on request from the corresponding author. The data are not publicly available due to confidentiality.

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
