# Peer review of "Exploring Chinese and Korean American Teachers’ Perceptions of Their Cultural Identity as Assets and Barriers"

_behavsci, 2023, doi:10.3390/bs13120969_

Round 1
Reviewer 1 Report
Comments and Suggestions for Authors
The manuscript entitled "Exploring Chinese and Korean American Teachers' Perceptions 2 of their Cultural Identity as Assets and Barriers" falls within the scope of the journal where it was submitted.
The abstract should better explain the methodology (typology) used, the methodological techniques and resources involved in order to draw certain conclusions that are put forward without proper detail.
The introduction and contextualization of the study are well explained and presented.
The structuring and presentation of the research problem and objectives are well done.
The sample should have been larger in order to be meaningful and to be able to gauge more consistent data and results for the study in general. If this is not possible, this limitation should be justified. Even to better understand and frame the issue of "representation emerged as a prominent theme that was divided into two sub-themes: Asian Representation (specific nuances to serve as representation for Asian students) and Asian x Female (specific nuances to serve as representation for Asian girls)". It would be important to better explain the methodological options and their possible influence on the final result of the research.
The use of zoom for the interview should be justified. Whether it was a methodological choice or whether it was due to convenience / opportunity or the impossibility of carrying out the interview in a more conventional way.
The limitations for future research are very carefully explained.
There is an obvious need to present a summary table of the main conclusions of the study so that the results and conclusions referred to by the authors can be graphically and/or visually understood from this study which, as mentioned above, has relevance and is important for discussion within the specific area under debate.
Reviewer 2 Report
Comments and Suggestions for Authors
This is an interesting topic and well-written paper. However, the authors could address some issues.
In the Method section, specifically, in point 2.3 Data analysis, a table could be added with the codes and their grouping.
In the Results section, the authors should review the textual quotes from the interviews, each contribution should be coded anonymously. In the current document no type of coding appears, for example (line 298-302; etc.). It could be put in a different font as well. In short quotes add the code of each person interviewed (e.g. line 306; line 315-317; line 322-323; etc.).
In the section Limitations and Future Research, we can also indicate that, given that we have a fairly small sample, it would be interesting to increase the number of interviews for future research. For example, it would be interesting to contrast more data with other types of participants, students, school management teams, families, etc.
Round 2
Reviewer 1 Report
Comments and Suggestions for Authors
After checking the responses, updates, additions and deletions, the manuscript is ready for publication. In short, all the questions raised have been clarified and justified and, in our opinion, the manuscript is more internally coherent and structurally complete.